# Aberrant activity of mitochondrial NCLX is linked to impaired synaptic transmission and is associated with mental retardation

Alexandra Stavsky[1,2,], Ohad Stoler[1,2,], Marko Kostic[1,2,], Tomer Katoshevsky[1,2,], Essam A. Assali [1,2,],
Ivana Savic[1,2,], Yael Amitai[1,2,], Holger Prokisch [3,4,], Steffen Leiz[5,], Cornelia Daumer-Haas[6,],
Ilya Fleidervish [1,2,], Fabiana Perocchi [7,8,], Daniel Gitler [1,2,9,✉] & Israel Sekler[1,2,9,✉]

Calcium dynamics control synaptic transmission. Calcium triggers synaptic vesicle fusion, determines release probability, modulates vesicle recycling, participates in long-term plasticity and regulates cellular metabolism. Mitochondria, the main source of cellular energy, serve as calcium signaling hubs. Mitochondrial calcium transients are primarily determined by the balance between calcium influx, mediated by the mitochondrial calcium uniporter (MCU), and calcium efflux through the sodium/lithium/calcium exchanger (NCLX). We identified a human recessive missense *SLC8B1* variant that impairs NCLX activity and is associated with severe mental retardation. On this basis, we examined the effect of deleting NCLX in mice on mitochondrial and synaptic calcium homeostasis, synaptic activity, and plasticity. Neuronal mitochondria exhibited basal calcium overload, membrane depolarization, and a reduction in the amplitude and rate of calcium influx and efflux. We observed smaller cytoplasmic calcium transients in the presynaptic terminals of NCLX-KO neurons, leading to a lower probability of release and weaker transmission. In agreement, synaptic facilitation in NCLX-KO hippocampal slices was enhanced. Importantly, deletion of NCLX abolished long term potentiation of Schaffer collateral synapses. Our results show that NCLX controls presynaptic calcium transients that are crucial for defining synaptic strength as well as short- and long-term plasticity, key elements of learning and memory processes.

[1] Department of Physiology and Cell Biology, Faculty of Health Sciences, Ben-Gurion University of the Negev, Beer-Sheva, Israel. [2] Zlotowski Center for Neuroscience, Ben-Gurion University of the Negev, Beer-Sheva, Israel. [3] Institute of Human Genetics, School of Medicine, Technische Universität München, Munich, Germany. [4] Institute of Neurogenomics, Helmholtz Zentrum München, Munich, Germany. [5] Department of Pediatrics, Klinikum Dritter Orden, Munich, Germany. [6] Pränatal-Medizin München, Munich, Germany. [7] Institute for Diabetes and Obesity, Helmholtz Diabetes Center (HDC), Helmholtz Zentrum München, Munich, Germany. [8] Munich Cluster for Systems Neurology, Munich, Germany. [9]These authors contributed equally: Daniel Gitler, Israel Sekler. ✉email: gitler@bgu.ac.il; sekler@bgu.ac.il

Calcium serves as a highly versatile intracellular signal that regulates a multitude of different cellular processes. In chemical synapses, calcium is the main driver of synaptic transmission; Neurotransmission is initiated in response to the influx of calcium ions through voltage gated calcium channels (VGCCs) in the presynaptic terminal, which is triggered by invading axonal action potentials[1–3]. There the ion binds to calcium sensors on the synaptic vesicles (SVs), inducing SV fusion with the presynaptic plasma membrane and secretion of their neurotransmitter content into the synaptic cleft[4–6]. In addition to being the initiator of synaptic transmission, calcium has a pivotal role in regulating SV endocytosis, mobilization, and recycling[7–9], and it also participates in short and long-term synaptic plasticity processes[10,11]. In order to exert tight control on synaptic activity and plasticity, neurons utilize efficient calcium buffering mechanisms that control its spatio-temporal distribution. The strategic localization of mitochondria in pre- and postsynaptic terminals[12–14] and their large capacity to sequester and release calcium ions[15,16] makes them ideally suited to regulate synaptic spatial and temporal calcium dynamics during neuronal activity. Consequently, mitochondria can regulate neurotransmission and synaptic plasticity[12,17].

Mitochondrial calcium transients are determined by a balance between calcium influx and efflux. Powered by the steep electrical potential across the inner membrane of the mitochondria ($\Delta\Psi_m$, −180 mV), cytosolic calcium flows rapidly into the matrix, primarily via the mitochondrial calcium uniporter (MCU)[18–21]. Calcium efflux is substantially slower (10-100 fold) and is mediated by the mitochondrial sodium/lithium/calcium exchanger (NCLX)[22,23], which is the main calcium extruder from mitochondria of excitable cells[24]. Due to its slower rate of transport compared to the MCU, it is the rate limiting step in defining mitochondrial calcium transients[25]. The physiological significance of the transients is substantial. They upregulate enzymes of the Krebs cycle[26–28], linking global changes in cellular calcium to increases in mitochondrial metabolic activity[29–32]. Furthermore, by controlling the local concentration of cytosolic calcium, mitochondria can strongly regulate the activity of plasma membrane and endoplasmic-reticulum calcium channels, which are allosterically regulated by calcium[16,33,34].

Imbalances in mitochondrial calcium influx/efflux/buffering can trigger mitochondrial calcium overload, which could lead to cell death. It has furthermore been linked to various pathologies, ranging from ischemia to neurodegenerative diseases[17,35,36]. Consistent with the unique rate-limiting role of NCLX, a conditional knock-out (KO) of NCLX in cardiac tissue led to mitochondrial calcium overload, triggering mitochondrial depolarization, which in turn reduced the magnitude of mitochondrial calcium transients[37]. While the impact of MCU KO on the cardiac system was surprisingly mild[38,39], deletion of NCLX in cardiomyocytes was lethal. In addition, a recent study revealed a reduction in the expression of NCLX in Alzheimer's disease patients[40]. These findings underscore the unique role of mitochondrial calcium efflux mediated by NCLX in health and disease.

Considering the important role of mitochondria in controlling synaptic calcium transients and energy provision, we examined the involvement of NCLX in regulating neuronal transmission. Our study was strongly motivated by our identification of a human NCLX variant linked to mental retardation, which we confirmed was unfunctional when expressed in a heterologous expression system. Consequently, we examined synaptic properties in primary hippocampal cultures and acute hippocampal slices of NCLX knock-out (NCLX-KO) mice. Our results indicate that NCLX is essential for preventing mitochondrial calcium overload, for executing mitochondrial calcium efflux and for maintaining the mitochondrial membrane potential. By affecting these parameters, NCLX controls presynaptic calcium levels, and consequently, the initial synaptic release probability and the magnitude of synaptic facilitation. Finally, our results indicate that NCLX is crucial for synaptic long-term potentiation. Thus, NCLX regulates critical aspects of mitochondrial function related to key neuronal properties.

## Results

**Mental and cardiac disabilities in a family harboring a P367S variant of NCLX.** Screening of WES data from undiagnosed patients with suspected mitochondrial disorders[41] identified a Pakistani family exhibiting a homozygous mutation in *SLC8B1* (the human *NCLX* gene) which is located on chromosome 12. The parents, who are first cousins, were heterozygous carriers of a c.1099 G > A variant (transcript uc001tvc.3) and 2 out of their 6 children were homozygous males for this substitution (Fig. 1a). The variant is predicted to cause a P367S substitution in a putative trans-membrane region of NCLX, in the vicinity of the catalytic domain (Fig. 1b). The P367 residue is evolutionary well-conserved (Fig. 1c), implying it is functionally important. Clinical assessment of both homozygous siblings revealed that they suffer from severe mental retardation and cardiomyopathies. To assess the functional significance of the P367S substitution on mitochondrial calcium extrusion, we expressed exogenous wild-type (WT) NCLX or the NCLX$^{P367S}$ variant in SH-SY5Y cells in which the endogenous NCLX had been knocked-down, and measured the kinetics of their mitochondrial calcium transients using Cepia2-mt, a mitochondrially targeted calcium sensor[42]. Expression levels of WT and P367S NCLX were similar as was their mitochondrial localization (Fig. 1d, Supplementary Fig. 3), implying that the P367S mutation does not affect protein expression or stability. Upon inducing an increase in cytoplasmic calcium by the application of ATP[43], mitochondrial calcium increased under both conditions (Fig. 1e). We observed that the efflux rate was significantly slower in cells expressing the NCLX$^{P367S}$ variant (Fig. 1f). Knock-down of endogenous NCLX also significantly slowed calcium extrusion (Supplementary Fig. 1a) as was previously shown[42,44]. Importantly, reintroduction of WT NCLX rescued calcium extrusion, while the P367S variant did not (Supplementary Fig. 1b), illustrating that the P367S NCLX variant is deficient in supporting calcium efflux. Because these results illustrate that loss of NCLX function is linked to severe cognitive impairment, we hypothesized that disruption of calcium extrusion from the mitochondria interferes with synaptic function and may alter mechanisms of learning and memory.

**Mitochondria in axons of NCLX KO neurons sequester at rest higher amounts of calcium and are partially depolarized.** Synapses are a key component of learning and memory processes[45,46], and recent studies underscore the influential role of mitochondria in them[47]. To experimentally explore how NCLX sculpts synaptic calcium handling and synaptic function, we used a NCLX knockout (NCLX-KO) mouse to approximate the loss of function of the NCLX$^{P367S}$ variant (Fig. 1). We examined free and buffered calcium in axonal mitochondria of primary hippocampal neuronal cultures from WT and NCLX-KO mice, by expressing the mitochondrially-targeted calcium sensor 2MT-mCherry-GCaMP6m (MitoRGCaMP) (Fig. 2a). In this construct, the GCaAMP6m green component is responsive to alterations in calcium levels whereas the red mCherry one is not. Consequently, the fluorescence intensity ratio (G/R) reports on calcium changes, factored for the expression level of the sensor and the dimensions of the mitochondria. MitoRGCaMP produced a punctate pattern in axons, and an elongated one in dendrites, consistent with the

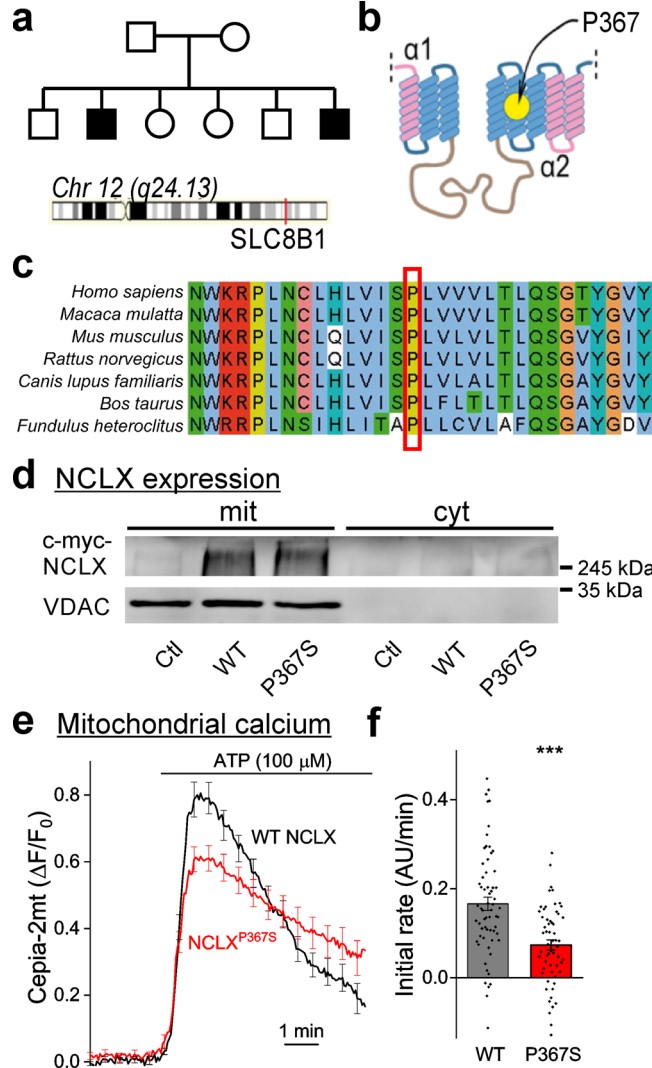

**Fig. 1 Genetic variation in human NCLX is linked to mental retardation and affects mitochondrial calcium efflux rates. a** Genealogical tree of the affected family. Affected retarded siblings are marked in black. The location of the *SLC8B1* gene in chromosome 12 is indicated. Both parents are carriers of the P367S variant of NCLX. **b** Predicted structure of NCLX. Yellow circle —location of P367 in the putative fifth transmembrane α-helix, adjacent to the catalytic site of NCLX. **c** P367 is phylogenetically conserved. Sequence alignment of a protein segment around P367 (red square) in various mammals and in fish illustrating its conservation across vertebrate species. **d** NCLX expression levels in HEK293T cells. Representative Western blot ($n = 3$ independent samples) of HEK cells expressing myc-WT NCLX (WT) or myc-NCLX$^{P367S}$ (P367S). mit – mitochondrial fraction (positive for the mitochondrial marker VDAC), cyt – cytoplasmic fraction (negative for VDAC). Anti myc immunoblotting revealed similar expression levels. Ctl – cells transfected with a control plasmid (pcDNA3.1 + ). **e** SH-SY5Y cells expressing NCLX$^{P367S}$ exhibit slower mitochondrial calcium efflux. Cepia2-mt and WT NCLX or NCLX$^{P367S}$ were co-expressed in SH-SY5Y cells in which the endogenous NCLX was knocked down by shRNA, and mitochondrial transients (Cepia2-mt fluorescence) were induced by bath application of ATP ($n = 62$ and $59$ WT and P367S traces, mean ± SEM $\Delta$F/F$_0$). **f** Quantification of calcium efflux rates in e. A linear fit of a 150 s period after calcium levels started to decline served to determine the initial calcium efflux rate ($0.166 \pm 0.016$, $0.073 \pm 0.011$ arbitrary units/minute in WT-NCLX and NCLX$^{P367S}$ expressing cells). ***$p = 8e-6$, Mann–Whitney u test ($Z = -4.45$, U = 990).

different morphologies of mitochondria in these compartments[48]. Furthermore, the overlapping pattern of GCaAMP6m and mCherry fluorescence matched that of the MitoView 405 dye, a live marker of mitochondria (Fig. 2a). We found that the G/R ratio was higher in the axonal mitochondria of resting NCLX-KO neurons (Fig. 2b), suggesting NCLX-KO neurons exhibit a higher basal level of free calcium in their mitochondria. This was not accompanied by differences in the mCherry component (Fig. 2c) implying the mitochondrial dimensions were unaffected. Indeed, quantification of the morphology of axonal mitochondria or their count per axonal segment did not differ between WT and in NCLX KO neurons (Supplementary Fig. 2). Most of the calcium in the mitochondria is bound to phosphate, and is thus termed "buffered mitochondrial calcium"[49]. We reasoned that an elevation in basal free mitochondrial calcium may also be reflected in the buffered pool. To examine this hypothesis, we treated cells expressing the construct MitoGCaMP6m with the uncoupling agent FCCP that disrupts the trans-mitochondrial pH gradient, thus releasing the buffered calcium[49,50]. When normalizing the initial fluorescence of the mitochondria by that measured after full depolarization, higher values were observed in NCLX-KO neurons (Fig. 2d), implying that mitochondria in resting NCLX-KO neurons exhibit higher levels of buffered calcium. The mitochondrial membrane potential ($\Delta\Psi_m$) is strongly affected by the resting mitochondrial calcium concentration and the calcium load[51,52]. We therefore assessed $\Delta\Psi_m$ using the potentiometric dye TMRM (tetramethylrhodamine methyl ester)[53] before and after depolarizing the mitochondria with FCCP. We found that in resting conditions, the mitochondrial membrane potential in NCLX-KO neurons was depolarized compared to WT ones (Fig. 2e), and calculated the difference to correspond to a depolarization of ~23 mV[44,54] (see "Materials and Methods"). Altogether, we show that mitochondria that lack NCLX sequester higher amounts of calcium at rest (free and buffered) and are partially depolarized, implying that NCLX plays a key role in shaping the mitochondrial resting calcium load and membrane potential, similarly to previous observations in cardiac cells[37].

A decrease in the mitochondrial membrane potential combined with calcium overload at rest, as we found in NCLX-KO neurons, could reduce mitochondrial calcium influx and/or efflux during neuronal activity, when calcium flows into the terminal through voltage gated calcium channels[33,55,56]. To explore this possibility, we field-stimulated hippocampal neurons expressing MitoRGCaMP at 20 Hz for 1 s (Fig. 2f). When considering the already-elevated calcium baseline, the amplitude of calcium influx into the mitochondria of NCLX KO neurons was smaller ($\Delta$G/R, Fig. 2g). The influx rate was slower in NCLX-KO (Fig. 2h), suggesting that either the MCU is downregulated[37,57] or the driving-force for calcium influx is decreased. In contrast, the rate of calcium extrusion was decreased in NCLX KO neurons compared to WT ones (Fig. 2i), showing that calcium extrusion is slower (see also Supplementary Fig. 1). These results underscore the key role of NCLX in controlling mitochondrial calcium dynamics in the axonal mitochondria.

**NCLX deletion affects cytoplasmic calcium levels at presynaptic boutons.** Mitochondria are present at a high incidence in presynaptic terminals[13] and can be also tethered to vesicle release sites[58]. Together with their ability to handle high calcium fluxes[59,60], they can have a substantial impact on the spatiotemporal distribution of calcium in the cytoplasm[15,61]. We therefore sought to monitor the dynamics of cytoplasmic calcium within presynaptic terminals during induced synaptic activity.

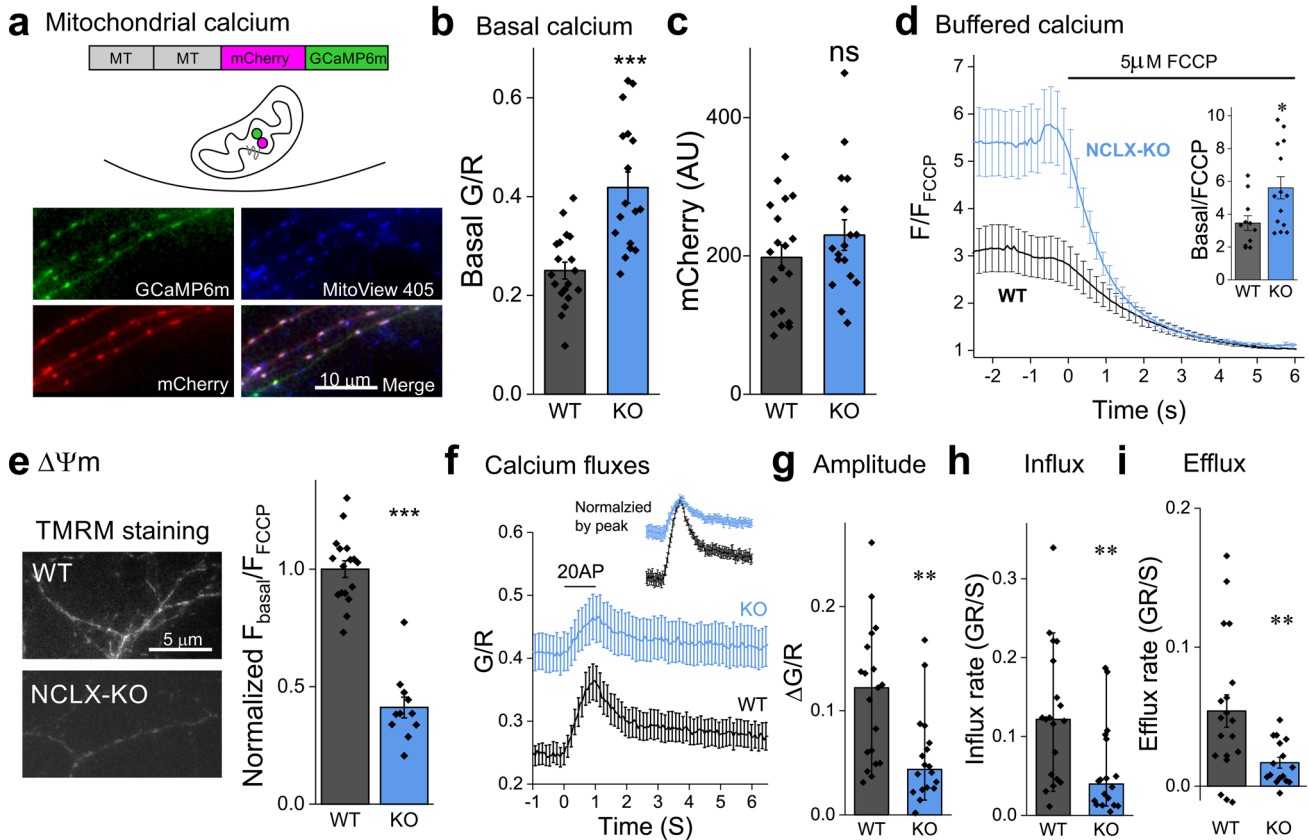

**Fig. 2 NCLX deletion increases resting mitochondrial calcium levels, depolarizes mitochondria and weakens calcium clearance during neuronal activity. a** Ratiometric measurement of mitochondrial calcium using MitoRGCaMP. MitoRGCaMP includes two mitochondrial targeting sequences (MT), calcium-insensitive mCherry and GcAMP6m, inducing localization of MitoRGCaMP to the mitochondrial matrix. Shown is an example of axons of WT primary hippocampal neurons expressing MitoRGCaMP to which the vital stain MitoView 405 was applied to label mitochondria, illustrating that the green and red components of MitoRGCaMP colocalize within the mitochondria. **b** Deletion of NCLX increases basal mitochondrial calcium levels. Quantification of the ratio between the fluorescence intensity of GCaMP6m (G) and that of mCherry (R) in the mitochondria of resting neurons in WT and NCLX-KO neurons (0.25 ± 0.02, 0.42 ± 0.03 mean ± SEM G/R), $n = 17$ and 19, respectively, ~50 mitochondria each, ***$p = 2e-5$, two-sided Student's $t$ test ($t = -4.832$, DF = 34). **c** Mitochondrial volume is unaffected by deletion of NCLX. Quantification of the fluorescence values of the calcium-insensitive mCherry component of MitoRGCaMP in the axonal mitochondria (as in (**b**)). No significant difference was observed between WT and NCLX-KO mitochondria (197.7 ± 18.3, 230.1 ± 22.2 mean ± SEM AU), $n = 17$ and 19, respectively, ~50 mitochondria each, ns $p = 0.27$, two-sided Student's $t$ test ($t = -1.133$, DF = 34). **d** Buffered calcium levels in axonal NCLX-KO mitochondria are lower. Depolarizing the mitochondria by bath application of FCCP (bar) leads to a leak of buffered mitochondrial calcium. FCCP was applied to neurons expressing MitoGCaMP6m in the absence of extracellular calcium to avoid mitochondrial calcium transients related to influx of calcium into the cytoplasm during plasma-membrane depolarization. Fluorescence intensity (F) per each mitochondrion was normalized by the value measured during the FCCP-induced plateau ($F_{FCCP}$). Inset: Ratio of fluorescence measured before and after the application of FCCP (3.46 ± 0.44, 5.61 ± 0.68 mean ± SEM $F/F_{FCCP}$), $n = 11$ and 14, respectively, >75 mitochondria each, *$p = 0.021$, two-sided Student's $t$ test ($t = -2.487$, DF = 23). **e** Mitochondria in NCLX-KO neurons are depolarized. Shown are representative fluorescence images of NCLX-KO and WT cultured hippocampal neurons bathed with the $\Delta\Psi_m$ indicator TMRM. FCCP was applied after baseline imaging to normalize the signal, as in (**d**). Initial $F/F_{FCCP}$ values were lower in NCLX KO neurons (1 ± 0.03, 0.41 ± 0.04 mean ± SEM, normalized by average WT value), $n = 17$ and 11, respectively, >25 mitochondria each, ***$p = 1e-10$, two-sided Student's $t$ test ($t = 10.35$, DF = 26). **f** Action-potential induced influx and efflux of calcium into mitochondria. Cultured hippocampal neurons expressing MitoRGCaMP were stimulated for 1 s at 20 Hz (bar) in the presence of APV and DNQX to block recurrent activity. Of note, the baseline value for the two genotypes is different (see (**b**)). Shown are mean ± SEM G/R traces. Image acquisition rate was 17.5 Hz. Inset: the same traces normalized by the peak of each trace. **g** Mitochondrial calcium transients in NCLX-KO neurons are smaller. $\Delta$G/R (peak- baseline) values, $n = 19$ and 18, respectively, ~50 mitochondria each, **$p = 0.0037$, Mann–Whitney test (U = 267, Z = 2.902), bars indicate medians and the error bars the 10–90 percentiles. **h** Calcium influx rate in NCLX-KO neurons is slower. Initial calcium influx rates, each calculated by performing a linear fit of the fluorescence values during the first 0.5 s of stimulation, $n = 19$ and 18, respectively, ~50 mitochondria each, **$p = 0.0032$, Mann–Whitney test (U = 266, Z = 2.872), bars indicate medians and the error bars the 10–90 percentiles. **i** Calcium efflux rate in NCLX-KO neurons is slower. Initial calcium efflux rates, each calculated by performing a linear fit of the fluorescence values during the first 0.5 s after cessation of stimulation (0.054 ± 0.012, 0.017 ± 0.004 mean ± SEM GR/S), $n = 19$ and 16, respectively, ~50 mitochondria each, **$p = 0.008$, two-sided Student's $t$ test ($t = -2.812$, DF = 33).

For that purpose, we used the SypI-mCherry-GCaMP6f (SyRG-CaMP) construct, which is localized to the presynaptic terminal by synaptophysin I, as evidenced by its co-localization with the endogenous vesicular protein vGlut1 (Fig. 3a). At rest, pre-synaptic G/R values were lower in NCLX-KO neurons (Fig. 3b, c), indicating lower basal calcium levels. Upon electrical stimulation (20 Hz, 1 s) we observed a lower net increase in calcium ($\Delta$G/R), reaching a lower G/R peak in NCLX-KO neurons, illustrating a lesser increase in presynaptic calcium during activity (Fig. 3b, d). The time constant of calcium clearance from the terminal was similar in NCLX-KO neurons (Fig. 3e), implying that this process is dominated by other calcium-handling mechanisms. Although a

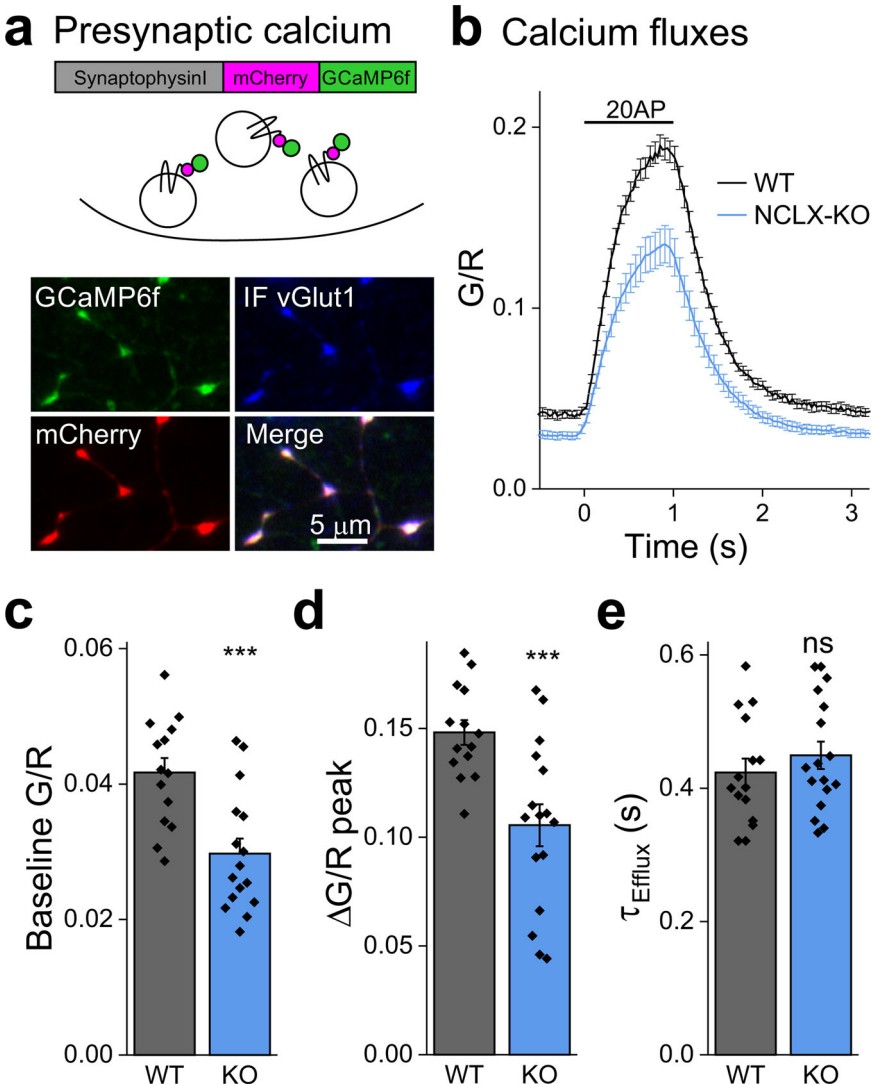

**Fig. 3 Basal and activity-induced cytoplasmic presynaptic calcium levels in WT and NCLX-KO neurons. a** Ratiometric measurement of cytoplasmic presynaptic calcium using SyRGCaMP. SyRGCaMP is constructed of synaptophysin I, mCherry (R) and GCaMP6f (G). Middle—SyRGCaMP is localized to synaptic vesicles. Shown is a representative image of synapses of WT primary neurons infected with SyRGCaMP and immunostained for vGlut1, a synaptic vesicle protein (blue), showing localization of SyRGCaMP on SVs. **b** Stimulation-induced synaptic calcium transients. WT and NCLX-KO neurons expressing SyRGCaMP were stimulated at 20 Hz for 1 s (bar). GCaMP6f fluorescence (G) was measured throughout and divided by mCherry fluorescence ratio (R), producing the (G/R) ratio. Shown are mean ± SEM G/R traces of $n = 14$ and 16 experiments, respectively, >55 boutons each. **c** Basal cytoplasmic calcium is lower in presynaptic terminals of NCLX-KO neurons. Distribution of mean basal G/R values calculated in 20 images acquired prior to stimulation, as in (**b**) (0.042 ± 0.002, 0.029 ± 0.002 mean ± SEM G/R), ***$p = 0.0006$, two-sided Student's $t$ test ($t = 3.852$, DF = 28). **d** Calcium transients are smaller in presynaptic terminals of NCLX-KO neurons. Increase in G/R values from baseline to peak, as in (**b**) (0.148 ± 0.006, 0.106 ± 0.010 mean ± SEM G/R), ***$p = 0.001$, two-sided Student's $t$ test ($t = 3.67$, DF = 28). **e** Calcium extrusion time constant unaffected by NCLX deletion. Time constants of the decay in G/R over time in (**b**), calculated by fitting a single exponent function to each trace (0.43 ± 0.02, 0.44 ± 0.02 mean ± SEM second), ns $p = 0.73$, two-sided Student's $t$ test ($t = -0.344$, DF = 28).

deficit in the capacity of mitochondria to cycle calcium could be expected to lead to higher cytoplasmic calcium transients[62], we observed the opposite, an observation which we address in the discussion section. Notwithstanding the specific mechanism, alterations in both basal and activity-dependent cytoplasmic calcium levels are expected to change synaptic properties; this prediction is interrogated below.

**Synaptic release is lower in NCLX-KO neurons due to a decrease in the probability of release.** Synaptic release is triggered by an increase in the calcium concentration in the presynaptic terminal[4,5]. To monitor synaptic function, we introduced synaptopHluorin (sypHy), a sensor of synaptic vesicle

recycling[63] into NCLX-KO and WT neurons. NCLX-KO neurons exhibited lower peak $\Delta F/F_0$ sypHy values during 5 s field stimulation at 20 Hz compared to WT neurons (Fig. 4a, b), suggesting that deletion of NCLX weakens synaptic release.

A possible explanation of this observation is that the attainment of lower calcium levels within the presynaptic terminal lowers the probability of release[64] ($P_r$). We therefore compared $P_r$ in glutamatergic synapses of the Schaffer collaterals of acute hippocampal slices of WT and NCLX-KO mice. To do so, we recorded NMDAR (N-methyl-d-aspartate receptor) responses to low-frequency stimulation of glutamate secretion, in a bathing medium lacking magnesium in the presence of MK-801, an activity-dependent irreversible NMDAR blocker and of

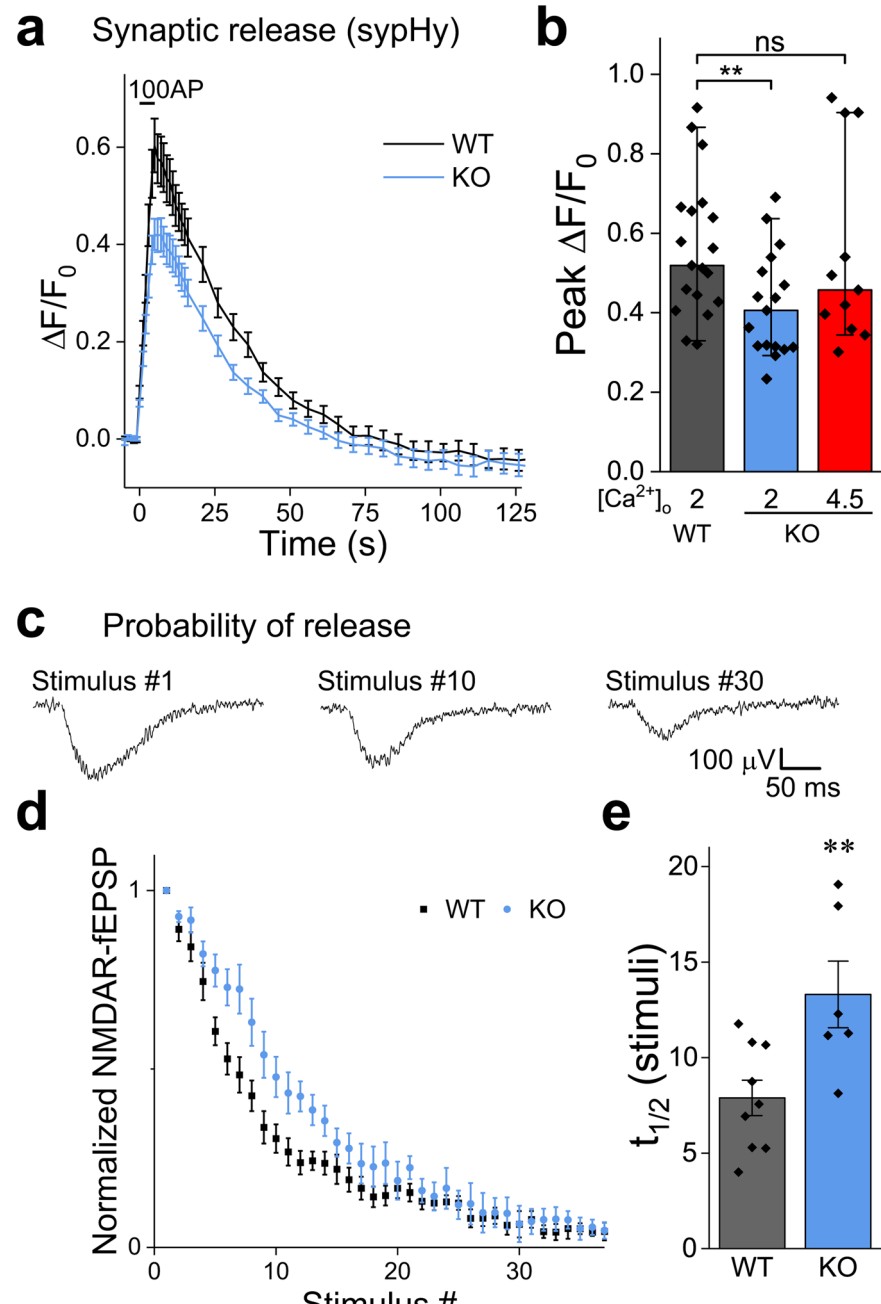

**Fig. 4 Deletion of NCLX weakens synaptic release and the lowers the initial release probability. a** SypHy responses are weaker in NCLX-KO neurons. WT and NCLX-KO neurons expressing sypHy were stimulated at 20 Hz for 5 s in 2 mM $Ca^{2+}$ saline. Shown are mean ± SEM sypHy traces. SypHy responses in the synapses of NCLX-KO neurons were lower. **b** Quantification of peak sypHy responses. Peak $\Delta F/F_0$ fluorescence, as in (**a**) and sypHy responses in NCLX-KO neurons bathed in $[Ca^{2+}]_o = 4.5$ mM shown in red, $n = 19$, 17 and 11, respectively, ~50 mitochondria each, $p = 0.04$ Kruskal-Wallis multiple comparison ANOVA (Chi-square = 6.42), post-hoc analysis using Mann–Whitney tests, **$p = 0.01$ (U = 243, Z = 2.57), ns $p = 0.52$ (U = 120, Z = 0.65), bars indicate medians and the error bars the 10–90 percentiles. **c** Progressive blockage of NMDARs by MK-801 used to assess $P_r$. The initial $P_r$ of Schaffer collateral synapses onto CA1 neurons was assessed by bathing slices in magnesium-free saline containing MK-801, DNQX and bicuculline. Stimulation was delivered every 15 S and NMDAR-fEPSPs were recorded. Shown are representative traces recorded from a brain slice of a WT mouse in response to the 1st, 10th and 30th stimuli, illustrating the progressive blockage of NMDARs by MK-801. **d** The initial synaptic probability of release (Pr) is lower in NCLX-KO neurons. NMDAR response amplitudes as a function of stimulus number, illustrating the different rate of progressive blockage of NMDAR-fEPSPs in WT and NCLX-KO neurons. mean ± SEM, $n = 9$ and 6 recordings from 4 WT and 3 NCLX-KO mice, respectively. **e** Half-life of NMDAR response blockage in (**d**) (7.89 ± 0.92, 13.31 ± 1.74, mean ± SEM in terms of stimulus number), **$p = 0.01$, two-sided Student's $t$ test ($t = -2.99$, DF = 13).

DNQX, to block AMPA receptors. This protocol leads to a progressive block of NMDAR responses, such that the rate of inhibition is proportional to $P_r$ (Fig. 4c;[65,66]). We found that blockage was slower in NCLX-KO slices (Fig. 4d, e), indicating that deletion of NCLX lowers $P_r$, consistent with both the lower

peak synaptic calcium and the smaller sypHy signal (Figs. 3, 4a). When $P_r$ was enhanced in NCLX KO slices by increasing the extracellular calcium concentration to 4.5 mM, peak synaptic release was similar to that observed in WT slices with 2 mM calcium (Fig. 4b), illustrating that the release capacity of

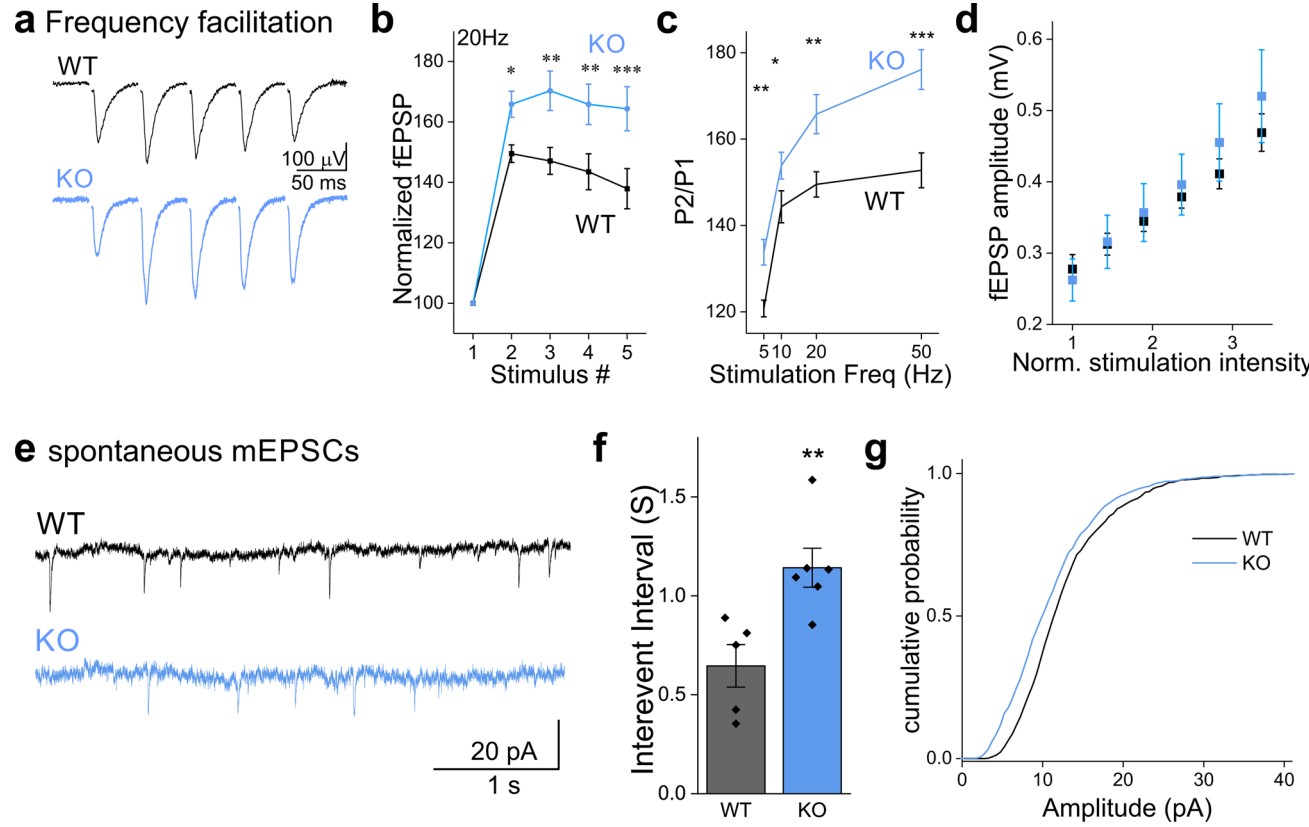

**Fig. 5 Synapses in NCLX-KO slices exhibit higher frequency-facilitation and more frequent spontaneous activity. a** Representative fEPSPs recorded from the CA1 area when delivering 5 stimuli to the Schaffer collaterals at 20 Hz in acute slices prepared from WT (black) and NCLX-KO (blue) mice (stimulation artifact was blanked). **b** Frequency-facilitation is higher in NCLX-KO slices. fEPSP amplitudes normalized by first response in each train (mean ± SEM), in WT and NCLX-KO slices stimulated at 20 Hz (n = 22 and 22 recordings from 8 WT and 9 NCLX-KO mice, respectively). The stimulation intensity in each train was set to produce an initial response of ~0.3 mV (see **d**). $p < 0.001$, two-way repeated-measures ANOVA ($F = 2452$, DF = 1,21) with Tukey's post-hoc analysis: $p = 0.029, 0.002, 0.003, 0.0006$ ($t = 3.15, 4.48, 4.31, 5.11$; DF = 63), respectively. **c** PPR is higher in NCLX-KO slices. Quantification of the P2/P1 ratio obtained at various stimulation frequencies. Stronger facilitation was observed in NCLX-KO slices at all frequencies. 5 Hz: $**p = 0.002$ two-sided Student's $t$ test ($n = 23,25$, $t = -3.24$, DF = 46); 10 Hz: $*p = 0.015$ ($n = 21,18$, $t = -2.549$, DF = 37); 20 Hz: $**p = 0.003$ ($n = 22,22$, $t = -3.14$, DF = 42); 50 Hz: $***p = 0.0005$ ($n = 19,18$, $t = -3.82$, DF = 35). **d** Response/input plot. The fEPSP response amplitude (±sem) is plotted as a function of the stimulation intensity ($n = 7$ WT, $n = 11$ NCLX KO). The stimulation intensity is normalized by the intensity in (mA) producing a response of ~0.3 mV in each slice. The graph illustrates that the responses are not saturated in both WT and NCLX-KO slices. **e** Intracellular recording of spontaneous mEPSCs. Representative traces recorded from WT (black) and NCLX-KO (blue) slices under resting conditions, in the presence of TTX. **f** Interevent intervals in NCLX-KO slices are longer. Interevent intervals were averaged in $n = 5$ and 6 slice recordings from 3 WT and NCLX-KO mice ($0.65 ± 0.11$, $1.14 ± 0.10$ mean ± SEM sec), $**p = 0.0079$, two-sided Student's $t$ test ($t = -3.397$, DF = 9). **g** mEPSC amplitudes are smaller in NCLX-KO slices. Cumulative plot of the mEPSC amplitudes recorded in WT and in NCLX-KO slices ($n = 990$ and 978 events, respectively). $***p < 0.001$, Kolmogorov–Smirnov test ($D = 0.163$, $Z = 3.61$).

NCLX-KO neurons is preserved. Thus, NCLX modulates synaptic strength by controlling synaptic calcium levels.

**Schaffer collateral pathway synapses from NCLX-KO mice exhibit stronger frequency facilitation.** To further support our conclusion that the $P_r$ is lower in NCLX slices (Fig. 4), we measured synaptic facilitation. Synaptic facilitation is a phenomenon in which postsynaptic potentials increase during repetitive stimulation, and is generally considered to result from the accumulation of residual calcium within the presynaptic terminal which leads to a progressive increase in the per-vesicle $P_r$[10,67,68]. We compared the relative change in fEPSPs amplitudes recorded in the CA1 area of the hippocampus during the delivery of 5 stimuli to the Schaffer collateral pathway in NCLX-KO and WT acute slices (Fig. 5a, b). In all stimulation frequencies we tested, NCLX-KO slices exhibited higher facilitation (Fig. 5c), consistent with a lower initial $P_r$. For single stimuli, fEPSP amplitude was linearly dependent on stimulation intensity (Fig. 5d), indicating

that recordings were not saturated at the range of stimulation intensity we employed. These results therefore support a role for NCLX in controlling synaptic calcium, and therefore in modulating short term synaptic plasticity.

**Frequency of spontaneous synaptic release is lower in NCLX-KO slices.** A reduction in resting calcium levels (Fig. 3b, c) could lead to a decrease in the frequency of spontaneous synaptic release, based on the latter's well-documented dependency on calcium levels[69–71]. We therefore performed patch-clamp recordings from pyramidal neurons in the CA1 in the hippocampus in acute slices from NCLX-KO and WT mice and recorded spontaneous miniature excitatory post-synaptic currents (mEPSCs) in the presence of tetrodotoxin (TTX) (Fig. 5e). The frequency of spontaneous mEPSCs was approximately two-fold lower in NCLX-KO slices (Fig. 5f). Analysis of the synaptic quantal size revealed a slightly but significant decrease in NCLX-KO slices (Fig. 5g), which is typically attributed to changes in

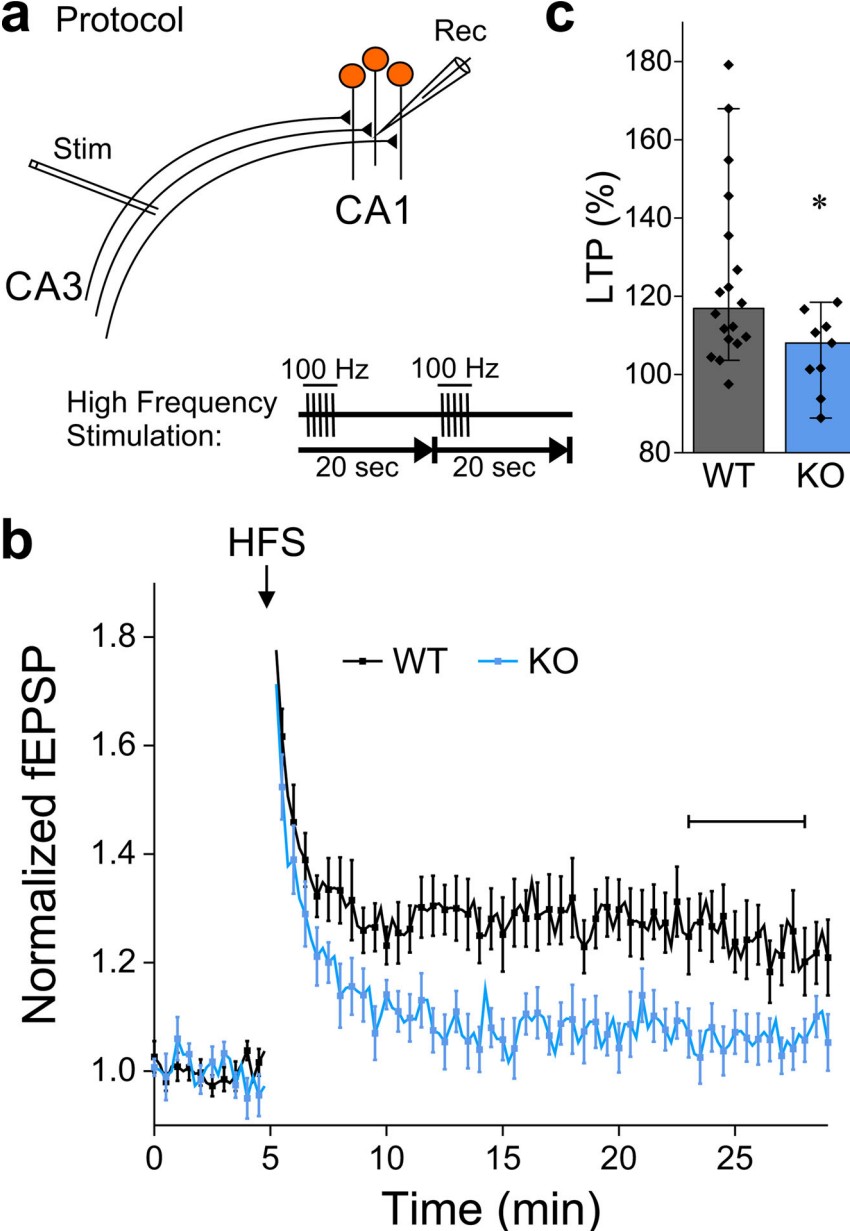

**Fig. 6 NCLX-KO slices fail to exhibit hippocampal Schaffer-collateral LTP. a** Schematic representation of extracellular recordings (fEPSP) from Schaffer collateral synapses in the CA1 area of acute hippocampal slices. The high-frequency stimulation (HFS) LTP induction protocol (5 pulses at 100 Hz, repeated once after 20 s) is shown below. **b** fEPSPs were evoked every 15 s, responses were normalized by baseline values. The CA3 area was stimulated to evoke fEPSP of ~0.3 mV. After 5 min of baseline recording, LTP was induced by HFS (arrow). Shown are mean ± SEM responses in WT ($n = 18$ slices from 7 mice) and NCLX-KO ($n = 9$ from 4 mice). **c** LTP in NCLX-KO slices is abolished. LTP during the horizontal bar in (**b**), $n = 18$ and 9 recordings, respectively, *$p = 0.029$, Mann–Whitney u test (U = 124, Z = 2.19), bars indicate medians and the error bars the 10–90 percentiles.

post-synaptic properties (see below,[72,73]), such as a lower density of receptors at the post-synaptic density or altered electrical properties of the postsynaptic neuron. Our results regarding short term plasticity and spontaneous synaptic activity are consistent with the lower basal calcium we observed in the presynaptic terminals of NCLX-KO neurons.

**Long-term potentiation is impaired in NCLX-KO slices.** Long-term potentiation (LTP) is a phenomenon in which brief tetanic stimulation of afferent fibers results in a long-lasting increase in synaptic strength or efficacy[74]. Calcium influx into postsynaptic terminals through NMDA receptors is necessary for generating and sustaining Schaffer collateral LTP[75–77]. Because our results

indicate that NCLX affects the calcium concentration at the presynaptic terminal during stimulation trains and that it affects neurotransmitter release, we hypothesized that it could also alter calcium dynamics at postsynaptic terminals, thus affecting LTP.

To test this hypothesis, we performed field recordings from the CA1 region of acute hippocampal slices of WT and NCLX-KO mice. LTP was induced by delivering high-frequency stimulation (HFS) to the Schaffer collaterals (Fig. 6a). In WT slices, following a brief period of post-tetanic potentiation (PTP), LTP was observed as a consistent increase in the magnitude of the postsynaptic response (Fig. 6b, c). In contrast, the responses in NCLX-KO slices decayed back to baseline, indicating that LTP induction failed (Fig. 6b, c).

## Discussion

We identified a loss of function variant of the human *SLC8B1* gene (NCLX[P367S]) in a Pakistani family in which the affected individuals exhibit severe mental retardation and cardiomyopathies, consistent with dysregulation of calcium dynamics in the brain and heart. Indeed, expression of NCLX[P367S] in SH-SY5Y cells, in parallel to knock-down of endogenous NCLX, led to substantial inhibition of efflux of mitochondrial calcium (Fig. 1). To examine in detail the neuronal consequences of loss of NCLX function, we measured synaptic calcium handling and synaptic transmission in neurons from NCLX knockout mice. We found that mitochondria of resting NCLX-KO neurons were overloaded and depolarized. Moreover, during neuronal activity, calcium influx into the mitochondria was reduced in parallel to a reduction in calcium efflux (Fig. 2). We also observed that cytoplasmic calcium transients were smaller in NCLX-KO neurons (Fig. 3). This can be explained by the following: (1) reduced cytosolic calcium transients lead to lower mitochondrial calcium influx. (2) The mitochondrial depolarization reduces the driving force for mitochondrial calcium influx, and (3) the rise in free mitochondrial calcium is known to inhibit further calcium influx by MCU[78]. Thus, our results indicate that deletion of NCLX in neurons affects not only mitochondrial calcium efflux (Fig. 2f, i), but also influx (Fig. 2f–h). We note that impaired mitochondrial calcium transport leads to inhibition of cell membrane calcium channels, amongst them the store operated calcium channels and VGCCs[33,56,79–81]. In agreement with the latter argument, NCLX-KO neurons exhibited weaker evoked synaptic transmission, lower *Pr* and stronger frequency-facilitation (Figs. 4 and 5). Interestingly, the amplitude of synaptic release could be compensated by elevating extracellular calcium (Fig. 4b), suggesting that the synaptic release machinery was not compromised; rather the decrease in *Pr* and synaptic release were consistent with the observed lower increase in synaptic calcium. Also, basal synaptic calcium in NCLX-KO neurons was also lower than in WT neurons (Fig. 3); in agreement, spontaneous synaptic activity in NCLX-KO slices was significantly less frequent (Fig. 5f). Finally, we found that SC-LTP, which is firmly linked to hippocampal-based learning and memory paradigms, was abolished in slices from NCLX-KO mice (Fig. 6). Our results lead to the intriguing hypothesis that the wide-range of observed synaptic impairments is the basis of the mental retardation observed in the siblings harboring the NCLX[S367P] variant, as has been observed for various mutations affecting synaptic functions[82,83]. Moreover, the observed decrease in the mEPSC amplitude (Fig. 5g) could indicate a compensatory decrease in the membrane-insertion of postsynaptic glutamate receptors, but alterations in the electrical properties (input resistance, dendritic tree size) of the postsynaptic cells could also contribute to this observation. Thus, the relative significance of primary synaptic derailments and of concurrent neurodevelopmental deficits remains to be explored.

Our results support a role for NCLX, and more broadly for mitochondrial calcium handling, in basic synaptic mechanisms and in the induction of LTP. We did not explore the mechanism by which NCLX deletion affects LTP; Nevertheless, we note that efflux of calcium that had accumulated in mitochondria during bouts of activity has been proposed to contribute to a sustained elevation in cytoplasmic calcium which contributes to the formation of LTP[84,85]. We postulate that in NCLX-KO neurons, overloaded mitochondria cannot regulate calcium levels efficiently and thus cannot contribute toward the formation of LTP. Indeed, calcium handling by mitochondria has been reported to affect synaptic properties; deletion or decreased expression of MCU, the mitochondrial calcium uniporter, leads to deficits in exocytosis or synaptic vesicle recycling[62,86], even though in both cases a significant increase in synaptic calcium was reported

during stimulation. Moreover, the mitochondrial exchanger has been proposed to participate in post-tetanic potentiation (PTP) in mossy-fiber synapses and in the calyx of Held[87,88]. However, these experiments were performed using CGP37157, which in addition to inhibiting NCLX, also affects plasma membrane $Ca^{2+}$ channels and the mitochondrial membrane potential[89,90]. We note that an alteration in the induction of LTP could arise due to changes in post-synaptic calcium-handling capabilities, as alluded to above, but could also be caused by modifications in intracellular signaling, receptor trafficking or other mechanisms that participate in LTP induction[91,92]. Another interesting aspect of mitochondrial function in this context is their role in supporting structural plasticity and protein synthesis after LTP induction[47]. Also, here we concentrated on neuronal deficits due to deletion of NCLX, it is also interesting to conjecture that mitochondrial dysfunction in glial cells, which also participate in neuronal plasticity processes[93–95], could contribute to the LTP deficits which we observed. To conclude, it would be of interest to determine if the impaired LTP that we encountered in NCLX KO mice is linked to impaired $Ca^{2+}$ hemostasis, partial mitochondrial depolarization or other factors.

We found striking similarities between our results and those reported for the cardiac system. We note that conventional deletion of NCLX, as was studied here, did not produce a fatal phenotype, unlike the conditional deletion of NCLX in the heart[37]. This could be due to the different developmental stages in which NCLX functionality is annulled in the two mouse lines. Indeed, conditional deletion did not affect young mice. We presume that in the NCLX-KO mice compensatory mechanisms are activated during early developmental stages. For example, we observed higher calcium buffering capacity in the NCLX-KO mitochondria in parallel to a decrease in their influx capacity. The latter effect is probably triggered by the partial depolarization of the mitochondria, which reduces the electrical driving force for calcium entry, but could also be the result of secondary inhibition of the MCU by a rise in mitochondrial calcium[78].

In the present work we concentrated on the role of NCLX in determining the calcium-handling properties of the mitochondria. We note that the impact of NCLX deletion on the mitochondrial contribution to cellular metabolism, synthetic pathways and the generation of reactive species still needs to be considered in future work. For example, we observed that mitochondria in NCLX KO neurons are depolarized by an estimated 23 mV (Fig. 2e). Such a depolarization is considerable, and in addition to reducing the driving force for calcium influx, may also partially reduce the efficiency of mitochondrial ATP production. However, it should be noted that polarization differences can occur within the same mitochondrial network[96] and in various tissues. The most notable example is brown fat tissue, where physiological depolarization can exceed 30 mV[97]. Another example is pancreatic β cells, where fatty acid metabolism can markedly depolarize the mitochondria[98]. Neurons have been shown to provide their ATP requirements by either mitochondrial respiration or glycolysis, as required[86,99]. Therefore, it is possible that the impact of depolarization of the mitochondria following NCLX deletion on calcium dynamics and ATP production is different. Furthermore, the influence of NCLX function on mitochondrially-dependent synthetic pathways is also of substantial interest.

To conclude, in the present study we show that genetic loss of function of NCLX is linked to mental retardation. The observed synaptic deficits in NCLX-KO neurons are consistent with mitochondrial involvement in the regulation of synaptic transmission, affecting both basic mechanisms of release, and synaptic plasticity phenomena that are key for learning and memory processes in the brain.

## Methods

**Patients**. The patients were presented for dysmorphological judgment related to complex developmental disturbance of unclear cause. They were born maturely after uncomplicated pregnancy. At the age of around 6 months a developmental retardation was first perceived. Free walking was observed at the age of 4 years with no verbal ability. The NCLX mutation was identified by screening of WES data from undiagnosed patients with suspected mitochondrial disorders[41] identified a homozygous mutation in *SLC8B1* (the human NCLX gene). Patient diagnosis and screen were carried according to approved Helsinki protocol, the ethical committee of the Technical University of Munich, informed consent was obtained from the parents.

**Mice**. NCLX knock-out (NCLX-KO; Slc8b1[em1J]) mice, back-crossed onto the C57BL6 background, were obtained from Jackson Laboratory (Bar Harbor, ME) and grown at the Ben-Gurion University mouse facility. C57BL6 wild-type (WT) controls were obtained from Harlan Laboratories (Ein Kerem, Israel). The quality of various commercially available anti-NCLX antibodies was not adequate to compare NCLX expression in WT and KO tissue (see also[100]). Therefore, we verified the existence of a 13 base-pair deletion predicted to produce a frame shift in the first exon of NCLX-KO mice intended to disrupt NCLX expression. mRNA was extracted from brains of WT and KO mice using EZ-10 DNAaway RNA mini-prep kit (Bio-basic). cDNAs were produced using a cDNA synthesis kit (PCR-Bio), and NCLX cDNA was amplified using the forward primer 5′-GCACTAGAGCAG CCAGCCCGTGAG-3′ and the reverse primer 5′-CCGTAAGCCTTGCTGAGCT GGAAACAC-3′. DNA sequencing of the region of the deletion site was performed using the forward primer, which hybridizes upstream of the predicted deletion site. Animals were treated in accordance to the guidelines of the Ben-Gurion University Institutional Committee for Ethical Care and Use of Animals in Research protocol IL-07-02-2019(C).

**Primary hippocampal cultures**. Primary hippocampal cultures from P0-P2 pups of either sex were performed essentially as described previously[101,102]. Briefly, postnatal day 0–2 pups were decapitated, the brains quickly removed, hippocampi were dissected, sliced manually, and kept on ice in Hank's Balanced Salt Solution (HBSS, Biological Industries) supplemented with 20 mM HEPES at pH 7.4. Hippocampus pieces were incubated for 20 min at room temperature (RT) in digestion solution consisting of 5 ml HBSS, $CaCl_2$ 1.5 mM, EDTA 0.5 mM and 100 units of papain (Worthington) activated with cysteine (Sigma-Aldrich). The brain fragments were then triturated gently two times. Cells were seeded at a density of 80,000–100,000 cells per well on 12 mm #1 glass coverslips coated with poly-D-Lysine (Sigma-Aldrich). Initially, cells were plated in plating medium consisting of Neurobasal-A medium supplemented with 2% B27, 2 mM Glutamax I (Thermo-Fisher Scientific), 5% defined FBS (Biological Industries) and 1 μg/ml gentamicin (Biological Industries). After 24 h, the plating medium was replaced with serum-free culture medium that consisted of Neurobasal-A, 2 mM Glutamax I and 2% B27. Cultures were maintained at 37 °C in a 5% $CO_2$ humidified incubator for 12–15 days prior to staining or imaging.

**Viral constructs**. Primary hippocampal cultures from WT and NCLX-KO mice were infected with adeno-associated viral particles (AAV) carrying cDNAs of proteins of interest or of fluorescent reporters[102]. cDNAs [2MT-mCherry-GCaMP6m (MitoRGcAMP), SynaptophysinI-mCherry-GCaMP6f (SyRGCaMP), 2MT-GCaMP6m (mitoGCaMP6m) and synaptophysinI-2XpHluorin (sypHy)] were subcloned by restriction/ligation (restriction enzymes and T4-ligase were from Fermentas/Thermo Scientific Life Science Research) into a plasmid containing adeno-associated virus 2 (AAV2) inverted terminal repeats flanking a cassette consisting of the neuronal-specific human synapsin 1 promoter[103] (hSyn), the woodchuck post-transcriptional regulatory element (WPRE) and the bovine growth hormone polyA signal. Viral particles were produced in HEK293T cells using both the pD1 and pD2 helper plasmids[104] which encode the rep/cap proteins of AAV1 and AAV2, respectively. Primary cultures of hippocampal neurons were infected at 5 DIV and incubated for at least 7 days before imaging. Virus titer was set to produce 75-90% infection efficiency. Mitochondrial localization of MitoRGCaMP was verified by vital labeling of the culture with 100 nM MitoView 405 dye (Biotium) to visualize mitochondria.

**Determination of the morphology of axonal mitochondria**. Sparse labeling of mitochondria of cultured hippocampal neurons, which is required to image separate individual neurons, was achieved by transfecting cultures with Lipofectamine 2000 (Thermo-Fisher Scientific) instead of using viral vectors. The cultures were transfected at 3 DIV, according to the manufacturer's instructions, except transfection was performed with 200 ng of mito-DsRed plasmid DNA and 0.2 μl of the lipofection reagent, each diluted in 40 μl Opti-MEM (Thermo-Fisher Scientific) and then mixed. Before transfection, 0.8 ml of the culture medium was removed from each well in a 24-well plate and kept. The transfection mix was added to the wells for 4 h. The well content was then replaced with the kept culture medium for further incubation and supplemented up to 1 ml. At 14 DIV, the cultures were fixed, mounted, and imaged using a X60 1.4NA oil-immersion objective (Nikon). Images containing axonal mitochondria were analyzed in ImageJ (Fiji) using a semi-automated machine-learning trainable WEKA segmentation tool[97,105]. Briefly, images were filtered to remove uneven background using a rolling ball algorithm with a diameter of 50 pixels. The WEKA classifier produced a probability map in which each pixel is assigned a confidence level as being mitochondrial. A binary image was formed by applying a uniform threshold on the classification certainty (at 20%), followed by segmentation into objects. Morphological properties of the objects were determined: area, perimeter, integrated density [(sum of intensity), circularity ($4\pi*area/perimeter^2$), roundness ($4*area$)/($\pi*long\_axis^2$)], and aspect ratio (long axis/short axis). Properties were averaged per picture, and means ± SEM were calculated across pictures. To count the number of mitochondria per unit length of axon, line profiles were manually drawn along axonal segments, and the number of mitochondria per segment were counted and normalized by the segment length. Counts were averaged per picture, and means ± SEM were calculated across pictures.

**Western blot**. HEK293T cells were grown on 60 mm culture plates and transfected as described. For protein determination in whole-cell lysates, cells were washed with ice-cold PBS and lysed for 30 min on ice with RIPA buffer containing: sodium pyrophosphate 10 mM, Tritone X-100 2%, NaCl 100 mM, 5 mM EDTA pH 7.4, 5 mM EGTA pH 7.4, deoxycholate 0.5%, NaF 25 mM, sodium-orthovanadate 1 mM. Lysates were centrifuged at 4 °C for 20 min at 12,000 rpm and supernatant collected. For determination of protein expression in mitochondria, cell lysates were fractionated[44,106]. Protein concentration in the samples was determined using Pierce BCA protein assay kit (Thermo Fisher Scientific) according to the manufacturer's protocol. Equal amounts of protein (20ug) from lysates were resolved by SDS-PAGE and transferred to a nitrocellulose membrane. Immunoblot analysis (anti myc – Abcam ab18185 and anti VDAC1 [N-18] – Santa Cruz sc-8828) was performed as described previously[23,44].

**Immunolabeling**. 12–14 DIV cultured neurons were fixed with 4% paraformaldehyde (EMS) in PBS for 10 min, rinsed with PBS, permeabilized with 0.1% Triton X-100 in PBS for 2 min and blocked with 5% powdered skim-milk (Sigma-Aldrich) in PBS for 1 h. The cover-slips were rinsed, incubated with the primary antibody (Goat polyclonal anti vGlut1, Synaptic Systems; 1:1000) for 1 h, rinsed, incubated with the secondary antibody (Donkey anti goat IgG, labeled with AlexaFluor 647, Abcam, 1:1000) for 1 h, rinsed, and mounted in immmount (Thermo Fisher Scientific). All steps were performed at RT.

**Fluorescence microscopy**. Fluorescence measurements of primary neuronal cultures were performed on a Nikon TiE inverted microscope driven by the NIS-elements software package (Nikon). The microscope was equipped with an Andor sCMOS camera (Oxford Instruments), a 40×0.75 NA Super Fluor objective, a 60 × 1.4 NA oil-immersion apochromatic objective (Nikon), a perfect-focus mechanism (Nikon), EGFP and Cy3 TE-series optical filter sets (Chroma) as well as BFP and Cy5 filter sets (Semrock).

Experiments on SH-SY5Y cells were performed on a IX73 inverted microscope (Olympus) equipped with pE-4000 LED light source and Retiga 600 CCD camera. Images were acquired through a 20 X/0.5 NA Zeiss Epiplan Neofluar objective using Olympus cellSens Dimension software.

**Neuronal culture field-stimulation**. Cultured neurons on coverslips were placed in a stimulation chamber (RC-49MFSH, Warner Instruments) and stimulated at an intensity of 10 V/cm using a stimulus isolation unit (SIU-102B, Warner Instruments). Stimulation duration and frequency was controlled by an isolated pulse stimulator (2100, A-M Systems). Neurons were imaged in standard extracellular saline containing (in mM): NaCl 150, KCl 3, Glucose 20, HEPES 10, $CaCl_2$ 2, $MgCl_2$ 3, pH adjusted to 7.35 with NaOH, 310 mOsm; unless indicated otherwise.

**Mitochondrial calcium imaging of SH-SY5Y cells**. SH-SY5Y cells (ATCC) were cultured and transfected 24 h later using the calcium-phosphate precipitation protocol[22]. The cells were co-transfected with the mitochondrial calcium reporter Cepia2-MT (Addgene plasmid # 58218), a pLKO.1 plasmid harboring validated shRNA targeted against the 3′UTR region of endogenous SLC8B1-NCLX (Sigma Aldrich, Mission shRNA TRC number 5045), and a plasmid to express human WT NCLX (Addgene #75216) or the NCLX[P367S] variant. Note that WT and P367S NCLX inserts are insensitive to the shRNA construct because they lack the 3′UTR of endogenous NCLX. The NCLX[P367S] variant was generated by site-directed mutagenesis using Phusion® Hot Start Flex DNA Polymerase (New England Biolabs) using the forward primer 5′ CATCTGGTTATCAGCTCCCTGGTTG TGGTC 3′ and reverse primer 5′ GACCACAACCAGGGAGCTGATAACCAGA TG 3′. The cells were first bathed in Ringer solution[44], which was replaced with calcium-free Ringer's solution supplemented with 100 μM ATP[23]. The resultant mitochondrial calcium signals were imaged (480 nm/535 nm excitation/emission). Measured values were normalized by the average baseline throughout ($\Delta F/F_0$) and were averaged across multiple cells in the field of view. The rate of calcium efflux was calculated by linearly fitting the change in the fluorescence after the peak for 150 s[23].

**Mitochondrial calcium imaging in neurons.** Neurons expressing MitoRGCaMP were imaged in the presence of 10 μM 6,7-Dinitroquinoxaline-2,3(1H,4H)-dione (DNQX; Sigma-Aldrich) and 50 μM DL-2-Amino-5-phosphonopentanoic acid (APV; Sigma-Aldrich) to block recurrent network activity. An image of the red channel (R; mCherry) was acquired prior to stimulation, which we verified in separate control experiments was unaffected by stimulation. mCherry fluorescence values for each mitochondrion were obtained from an area of 3 × 3 pixels at its center of mass. During stimulation, GCaMP6m (G) was imaged at an acquisition rate of ~17 Hz (57 ms per image) at 3 × 3 binning. GCaMP6m fluorescence was measured in the same area as mCherry. The fluorescence intensity measured in adjacent axonal segments was subtracted from each channel, and a G/R ratio was calculated for each mitochondrion. 20 images acquired before stimulation served to obtain the basal G/R ratio. A mean trace for the specified number of mitochondria was calculated for each experiment, and traces were averaged across independent experiments for each condition (from >3 independent cultures). The peak amplitude was calculated as the G/R ratio at the peak minus the basal level. Initial rates of calcium influx and efflux (ΔG/R over time) are the average slope of linear fits through the first 10 data points of each trace, starting with initiation and cessation of stimulation, respectively (a segment of approximately 0.5 s in duration).

Release of buffered mitochondrial calcium in neuronal cells was performed in calcium-free extracellular solution containing (in mM): 150 NaCl, 3 KCl, 10 HEPES, 4 MgCl$_2$ and 20 glucose (pH adjusted to 7.35 with NaOH). Cells were imaged before and after bath application of 5 μM Carbonyl cyanide-p-trifluoromethoxyphenylhydrazone (FCCP; Santa Cruz Biotechnology) to induce full depolarization of the mitochondria. Fluorescence was normalized by the value obtained after full depolarization by FCCP ($F/F_{FCCP}$).

**Mitochondrial membrane potential measurements.** Tetramethylrhodamine Methyl Ester (TMRM; Invitrogen) was used in non-quenching mode to measure pre-existing ΔΨm. Cells were loaded with 25 nM TMRM for 15 min at RT while gently shaking. For imaging of basal TMRM fluorescence, cells were transferred into standard extracellular solution supplemented with TMRM, APV and DNQX. After imaging baseline levels, 5 μM FCCP was added to trigger full depolarization, enabling calibration of the measurement. Basal fluorescence was normalized to the fluorescence after FCCP application ($F_{FCCP}$) in each experiment. The results were normalized by the average $F/F_{FCCP}$ ratio measured in WT neurons. Changes in mitochondrial membrane potentials were calculated as follows[54,107]:

$$\triangle \Psi_x = 58.7 \log \frac{(F_{KO} - F_{KO-FCCP})}{(F_{WT} - F_{WT-FCCP})} \quad (1)$$

where ΔΨx is the difference in ΔΨm (in millivolts) between the control and KO condition; $F_{KO}$ is the basal TMRM fluorescence in NCLX-KO neurons; $F_{KO-FCCP}$ is the fluorescence in NCLX-KO neurons after FCCP application (the same for WT neurons).

**Presynaptic cytoplasmic calcium measurements.** Neurons infected with SyRGCaMP were imaged in the presence of APV and DNQX to block recurrent network activity. An image of the red channel (R; mCherry) was acquired before stimulation. We verified in separate control experiments that mCherry fluorescence is unaffected by stimulation. mCherry fluorescence values for each synapse were measured in a 3 × 3 pixels area at the center of mass of each synapse. During the stimulation phase, GCaMP6f (G) was imaged at an acquisition rate of 37 Hz (27 ms per image) using 3 × 3 binning. The G/R ratio was calculated for each synapse in each image in the time-lapse sequence as the background-subtracted fluorescence of GCaMP6f divided by the background-subtracted fluorescence of mCherry. For each experiment, the values recorded for the number of specified synapses was averaged, and then a mean value was calculated for the number of specific independent experiments per each condition (from >3 independent cultures).

**SyPhy: measuring vesicle cycling.** SypHy is a probe based on the internal fusion of pHluorin, a pH-sensitive GFP (pKa=7.6) with the vesicular protein Synaptophysin I (SypI), so that pHluorin is located in the lumen of the SVs, and allows the reportage of presynaptic activity[63]. SypHy is quenched by the acidic pH of intact vesicles (pH~5.5). Upon exocytosis, sypHy is unquenched by its exposure to the extracellular environment (pH~7.3). After endocytosis and reacidification, sypHy is requenched. An increase in fluorescence reflects exocytosis offset by concurrent endocytosis, and the subsequent decrement indicates endocytosis. In total, 12–14 DIV neurons were stimulated in the presence of APV and DNQX. Experiments were conducted in standard extracellular solution or in high calcium solution with 4.5 mM CaCl$_2$ and 0.5 mM MgCl$_2$. At the completion of each experiment, the bath was perfused with saline in which 50 mM NaCl was replaced with NH$_4$Cl to visualize the total vesicle population. Fluorescence was monitored in a time-lapse mode (20 images at 1 Hz and then 26 images at 0.2 Hz). The baseline fluorescence intensity of sypHy ($F_0$) in each synapse is the average of the values measured in 5 images acquired before stimulation. The change in fluorescence (ΔF) at time t was calculated as F(t)-$F_0$. For each experiment, the values recorded for the number of specified synapses was averaged, and then a mean value was calculated for the number of specific independent experiments per each condition (from >3 independent cultures).

**Acute hippocampal slices.** P18-P21 mice from either sex were anesthetized with isoflurane and decapitated. The brains were rapidly removed and placed in ice-cold oxygenated cutting solution that contains (in mM): 252 Sucrose, 5 KCl, 1 CaCl$_2$, 3 MgSO$_4$, 26 NaHCO$_3$, 1.25 NaH$_2$PO$_4$ and 10 Glucose; pH 7.3 when bubbled with 95% O$_2$/CO$_2$. Transverse slices (300 μm) were cut on a vibratome (Leica 3000) and placed into a holding chamber containing oxygenated ACSF at room temperature for at least an hour prior to the recordings. ACSF solution contains (in mM): 124 NaCl, 3 KCl, 2 CaCl$_2$, 2 MgSO$_4$, 1.25 NaH$_2$PO$_4$, 26 NaHCO$_3$, and 10 glucose; pH 7.4 when bubbled with 95% O$_2$/CO$_2$.

**Field-potential recordings.** Extracellular stimulation was delivered using a stimulus isolation unit (A.M.P.I.) using glass monopolar electrodes (0.5–1 MΩ) filled with ACSF. fEPSPs were recorded in current-clamp mode with a MultiClamp 700B amplifier (Molecular Devices) driven by the Clampex (pClamp 10.0) software (Molecular Devices) using ACSF-filled patch pipettes (0.5–1 MΩ). The stimulating electrode was placed in the CA3 area and the recording electrode in the dendritic CA1 area. Experiments were performed at 29 ± 1 °C with flow rates of 2 ml per minute. 5 consecutive stimulations were delivered at 5, 10, 20, and 50 Hz. Stimulation intensity was set to acquire an initial fEPSP of approximately 0.3 mV. In total, 5–10 sweeps were conducted for each stimulus frequency and recordings were averaged over trials. Data were sampled at 10 kHz, amplified (gain 5), filtered at 3 kHz then digitized and analyzed using Clampfit. Data were included if the stimulation intensity to fEPSP ratio was within the linear range and if there was an increase in the amplitude of the second response. Data were averaged per slice, and then across the number of specified slices. Up to 3 slices were usable per each mouse.

*Probability of release.* To record NMDAR-fEPSPs, magnesium was excluded from ACSF to relieve the Mg$^{2+}$ block of NMDA receptors. Slices were incubated for at least 30 min in magnesium-free ACSF containing (+)-Bicuculine (100 μM; Alomone Labs) to block inhibitory inputs and DNQX (5 μM) to block AMPA receptor-mediated responses. Stimulation was conducted every 15 s for 5 min to obtain a baseline response. Stimulation was halted for 10 min while 40 μM (+)-MK801 (Millipore) was added and allowed to equilibrate. Recordings then were conducted at the same rate for 6 additional minutes until 90% of the NMDAR response was inhibited.

Responses were renormalized so that the baseline was set to 1 and the final fEPSP amplitude was set to 0. The rate of fEPSP (NMDA) decline was quantified by calculating the t½ of the decay traces. Recordings that exhibited epileptiform activity were excluded from analysis.

**Long-term plasticity induction via high-frequency stimulation.** Responses to stimulation were recorded every 15 s for 15 min prior to the induction of long-term potentiation (LTP). LTP was induced by the delivery of high-frequency stimulation (HFS, 5 pulses at 100 Hz, repeated again after 20 s). Following induction, responses were further recorded for 30 min; the degree of potentiation was calculated as the ratio of the average fEPSP amplitude recorded 20–25 min after delivery of the HFS and the baseline amplitude. Traces were excluded from the analysis if the baseline was not stable.

**Whole-cell recordings.** Slices were viewed through 40X or 60X water-immersion lenses (Olympus) in a BX51WI microscope (Olympus) mounted on an X–Y translation stage (Luigs and Neumann). Somatic whole-cell recordings were made using patch pipettes pulled from thick-walled borosilicate glass capillaries (1.5-mm outer diameter; Science Products). Pipettes had resistances of 5–7 MΩ when filled with cesium-based intracellular solution (in mM: 135 CsCl, 4 NaCl, 2 MgCl2 and 10 HEPES (cesium salt), pH adjusted to 7.3 with CsOH, intended to extend the range of the space clamp). Voltage-clamp recordings (−70 mV holding potential) were made with a MultiClamp 700B amplifier (Molecular Devices). The extracellular ACSF solution was supplemented with 1 μM tetrodotoxin (TTX) and 10 μM Bicuculline methiodide (both Alomone labs, Israel) to block network activity and GABAergic transmission, respectively. Data was low-pass–filtered at 1 kHz (−3 dB, single-pole Bessel filter) and digitized at 10 kHz. Membrane access resistance was maintained as low as possible (5–10 MΩ) and was compensated at 80%. Recordings were not corrected for liquid junction potential. Recordings were performed at 30 °C. Inter-event intervals were calculated per recording (per slice) and averaged across slices (up to 3 slices per mouse).

**Statistics and reproducibility.** In all graphs, unless indicated otherwise, symbols/bars and error bars denote mean ± SEM. Sample sizes are reported in the figure legends. For neuronal imaging, mean ± SEM values were calculated across n images, and for each image an average value was calculated from multiple synapses/mitochondria (as indicated). Images were obtained from at least 3 independent cultures. For SH-SY5Y cells, mean ± SEM were calculated across individual cells, measured in at least 3 independent experiments. In slice recordings, n denotes slices, with up to 3 slices obtained from individual mice. For intracellular recordings, values were averaged across cells, recorded in slices from at least 3 mice per genotype. Testing for normality of distribution was performed using the Shapiro-Wilk normality test. Normally distributed pairs of datasets were

compared using the two-sided Student's $t$ test, or the Mann–Whitney's non-parametric u test otherwise. Comparison of multiple datasets that were not normally distributed was performed using the Kruskal-Wallis test, with u tests used for subsequent pairwise post-hoc analyses. Data obtained in a repeated-measures design were analyzed using two-way repeated measures ANOVA, using Tukey's post-hoc analysis. Single outliers were identified using Grubbs's test. Significance was set at a confidence level of 0.05 for all tests. In all figures, "ns" denotes $p >= 0.05$, * denotes $p < 0.05$, **$p < 0.01$, and ***$p < 0.001$. Statistical analysis was performed using Origin 2016/2020 (Originlab, Northampton MA).

**Reporting summary**. Further information on research design is available in the Nature Research Reporting Summary linked to this article.

## Data availability
The datasets generated during and/or analyzed during the current study are available from the corresponding authors on reasonable request.

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

## Acknowledgements

The project was funded by the BMBF through the German network for mitochondrial disorders, mitoNET (01GM1996B) (HP), ISF grants 1427/12 and 1310/19 (DG), ISF grant 1929/17 and DIP SE 2872/1-1 (ISe); the Munich Center for Systems Neurology (SyNergy EXC 2145 /ID 390857198), the ExNet-0041-Phase2-3 ("SyNergy-HMGU") through the Initiative and Network Fund of the Helmholtz Association and the Bert L & N Kuggie Vallee Foundation (FP).

## Author contributions

A.S., O.S., M.K., T.K., E.A.A., I.Sa., S.L., C.D.H., F.P., D.G. collected and analyzed data. I.Se., D.G. conceived the project. A.S., Y.A., I.F., H.P., D.G., I.Se. designed experiments. A.S., I.Se., D.G. wrote and edited the manuscript. All authors approved the manuscript.

## Competing interests

The authors declare no competing interests.
