## [Peer Review File · Communications Biology]

Reviewers' comments:

Reviewer #1 (Remarks to the Author):

In their manuscript, Stavsky and colleagues identify a missense mutation in the *Slc8b1* gene, which encodes the mitochondrial sodium/lithium/calcium exchanger NCLX, in individuals with severe mental retardation. In subsequent functional analyzes the authors show that the identified P367S is a LoF mutation. Neurons deficient in NCLX show impaired mitochondrial calcium regulation, and synaptic defects characterized by weakened synaptic release, higher frequency-facilitation and spontaneous activity and impaired LTP. The authors conclude that LoF of NCLX is (causally) linked to mental retardation. The manuscript is well written and the experiments and used methods are of high standard. The results and conclusions should be of interest to others in the community. The following points should be addressed:

- 1) While the results in Figure 1E show convincingly that P367S is a LoF mutation, it would be helpful to add mitochondrial calcium dynamics from NCLX KD cells to estimate if the P367S variant has any residual activity.
- 2) In Fig 1D, the authors should add a Western blot that shows endogenous NCLX levels and the knockdown efficiency upon shRNA transfection. Also, I could not find information about the shRNA knock-down in the material/methods section.
- 3) Does NCLX deficiency impact general mitochondria biology in neurons, e.g. morphology, respiratory chain supercomplex function, distribution of mitochondria in neurons?
- 4) Since the manuscript starts with the identification of the P367S variant in patients, some functional experiments should be performed in NCLX-KO neurons that express a P367S vs a NCLX wt construct.
- 5) Synaptic defects in NCLX-KO neurons are only characterized for excitatory synapses. However, mitochondria are also present at and critical for GABAergic synapses. Electrophysiology could be used to determine whether NCLX deficiency also impacts inhibitory synapses.
- 6) The authors show nicely that hippocampal Schaffer-collateral LTP is impaired. They could elaborate a bit more (experimentally or in the discussion) the potential molecular mechanism(s) at the post-synapse.
- 7) Based on gene expression, NCLX seems to be expressed much higher in astrocytes and microglia compared to neurons. While the presented neuronal defects are convincing, in primary neuronal cultures astrocytes are present in huge numbers. NCLX deficiency has been shown previously to have an impact on diverse astrocyte functions. Can the authors comment if the measured effects in neurons might be secondary to impaired functions of other cells, e.g. glia, that are present in the cultures/brain slices? Sparse knockdown of NCLX in neurons, like used in Fig. 1, could be used to show cell autonomous effects of NCLX deficiency in neurons.

Reviewer #2 (Remarks to the Author):

In the manuscript entitled "Aberrant activity of mitochondrial NCLX is linked to impaired synaptic transmission and is associated with mental retardation" by Alexandra Stavsky and colleagues describe neuronal alterations resulting from a disease-related mutation in the mitochondrial sodium/lithium/calcium exchanger (NCLX). The main finding is that the P367S mutation of NCLX leads to slowed calcium efflux from mitochondria which results in calcium accumulation in mitochondria and defective buffering of axonal calcium. As consequence, there are reduced basal levels and activity-dependent increase of calcium in neurons. Ultimately, these alterations cause reduced evoked and spontaneous release of neurotransmitter (glutamate) and impair the induction of LTP. I want to congratulate the authors for a well written article which I enjoyed reading, a very detailed method section (uncommon nowadays but necessary for reproducibility) and overall and interesting manuscript. I think that the data presented in the figures, using different techniques (molecular biology, electrophysiology, imaging) and experimental settings (cultures and slices), fully support the authors' conclusions and no further experiments are needed. I only have minor comments that require changes/additions in the text.

1. In the Discussion section the authors state that the “results indicate that deletion of NCLX affects not only mitochondrial calcium efflux, but also influx”. However, in Figure 1E calcium influx into the mitochondria seems to be similar between WT and NCLX-P367S cells, contradicting the previously mentioned conclusion. How do the authors reconcile this? I think there is no direct evidence in the manuscript for calcium influx alterations.

2. Regarding the changes in neurotransmission, the authors show a decrease in neurotransmitter release (using pHluorin and also estimating PR via electrophysiology). There is also a decrease in the amplitude of miniature postsynaptic currents, indicating putative changes in postsynaptic receptors but this is not discussed by the authors, can the authors speculate on the meaning of this result? Are postsynaptic receptors reduced in NCLX-P367S neurons? As a side note, there is no indication in the manuscript if the mEPSC recorded are AMPAR- or NMDAR-mediated (the traces look like AMPAR-mediated), this has to be informed.

3. Finally, the authors center the discussion on calcium regulation by mitochondria and its impact on neurotransmission. However, the observed calcium accumulation in mitochondria, taken together with the depolarization of the mitochondrial membrane, can also lead to impairment on ATP production and also generation of reactive species (and the consequent oxidative stress), both processes that can negatively impact on neuronal health and neurotransmission. These alternative explanations need to be added and discussed in the manuscript. As a small note related to this topic, the authors calculate a depolarization of the mitochondrial membrane of ~ 23 mV, is this a significant change? The meaning of this magnitude (i.e. if it is a big enough change to alter mitochondria metabolism) should be explained.

We would first like to thank the reviewers for comprehensively and carefully reading our manuscript and for their constructive and professional comments. We have addressed most of the comments, and believe that as a result the manuscript has been improved significantly and is now suitable for publication in Communications Biology. Following is a point by point answer to the reviewers' comments.

Reviewer 1

In their manuscript, Stavsky and colleagues identify a missense mutation in the Slc8b1 gene, which encodes the mitochondrial sodium/lithium/calcium exchanger NCLX, in individuals with severe mental retardation. In subsequent functional analyzes the authors show that the identified P367S is a LoF mutation. Neurons deficient in NCLX show impaired mitochondrial calcium regulation, and synaptic defects characterized by weakened synaptic release, higher frequency-facilitation and spontaneous activity and impaired LTP. The authors conclude that LoF of NCLX is (causally) linked to mental retardation. The manuscript is well written and the experiments and used methods are of high standard.

We thank the reviewer for the positive opinion of our manuscript.

1) While the results in Figure 1E show convincingly that P367S is a LoF mutation, it would be helpful to add mitochondrial calcium dynamics from NCLX KD cells to estimate if the P367S variant has any residual activity.

We have added supplemental figure 1 in which we show the effect of knocking down NCLX. These results show that while expression of WT NCLX on the background of the knock-down of endogenous NCLX supports calcium extrusion from the mitochondria, the P367S NCLX variant does not, as the reviewer suggested. We refer to this information in p.13-14.

2) In Fig 1D, the authors should add a Western blot that shows endogenous NCLX levels and the knockdown efficiency upon shRNA transfection. Also, I could not find information about the shRNA knock-down in the material/methods section.

Indeed, we mistakenly omitted the technical details concerning the shRNA procedure. This information has been added to the materials and methods section (under "Mitochondrial calcium imaging SH-SY5Y cells"). We used a validated commercial shRNA plasmid (from Sigma-Aldrich), the use of which we have already reported in earlier publications (supplemental figure 2 in both Kostic et al., 2015, 2018). Therefore, we refer the readers to our earlier publications in regards to the performance of the shRNA construct (p. 13). These previous publications include Western blots obtained using the same shRNA construct, illustrating its efficiency. Unfortunately, the antibody that was used in our previous studies is no longer available, and we have not been able to identify commercially available antibodies that successfully identify endogenous NCLX (see also Pathak et al. 2020 in relation to this issue, also mentioned in p. 4).

As indicated above, supplemental figure 1 illustrates that KD of NCLX significantly slows calcium extrusion from the mitochondria of SH-SY5Y cells. We are confident that the effect of the shRNA construct in this functional assay provides a clear answer also to this question.

Does NCLX deficiency impact general mitochondria biology in neurons, e.g. morphology, respiratory chain supercomplex function, distribution of mitochondria in neurons?

To answer the reviewer's questions, we added supplementary figure 2, in which we show that the morphology of axonal mitochondria and their number per axonal length are unaltered. We regret that due to current limitations (Our Seahorse is malfunctioning and cannot be repaired in these trying times), we are unable to address the question regarding the effects on the respiratory chain. However, notice that in answer to a question by reviewer 2, we added clarification of the effect of mitochondrial depolarization, which we did document, on mitochondrial respiration (see. p. 20-21). Certainly, this issue will need to be addressed directly in future publications.

4) Since the manuscript starts with the identification of the P367S variant in patients, some functional experiments should be performed in NCLX-KO neurons that express a P367S vs a NCLX wt construct.

We thank the reviewer for this excellent idea. [Redacted]

5) Synaptic defects in NCLX-KO neurons are only characterized for excitatory synapses. However, mitochondria are also present at and critical for GABAergic synapses. Electrophysiology could be used to determine whether NCLX deficiency also impacts inhibitory synapses.

We completely agree with the reviewer that mitochondria in GABAergic neurons are of significant interest, especially considering their high levels of activity (as some of them are fast spikers). However, we feel that this point is outside the scope of the present manuscript. We will certainly consider performing a specific study about the role of NCLX in GABAergic neurons. We thank the reviewer for this excellent suggestion.

6) The authors show nicely that hippocampal Schaffer-collateral LTP is impaired. They could elaborate a bit more (experimentally or in the discussion) the potential molecular mechanism(s) at the post-synapse.

As per the reviewer's request, we have expanded the discussion of the possible molecular mechanisms at the post-synapse (top of p. 20), also in answer to comments by reviewer #2.

Based on gene expression, NCLX seems to be expressed much higher in astrocytes and microglia compared to neurons. While the presented neuronal defects are convincing, in primary neuronal cultures astrocytes are present in huge numbers. NCLX deficiency has been shown previously to have an impact on diverse astrocyte functions. Can the authors comment if the measured effects in neurons might be secondary to impaired functions of other cells, e.g. glia, that are present in the cultures/brain slices? Sparse knockdown of NCLX in neurons, like used in Fig. 1, could be used to show cell autonomous effects of NCLX deficiency in neurons.

Indeed, brain function is certainly dependent also on the proper function of cells other than neurons, and thus the reviewer is correct in pointing out that deficits in mitochondrial function in glial cells could contribute towards the effects that we have observed in this manuscript, especially in respect to our LTP results, which were obtained in tissue slices. This being the case, we now specifically mention in the discussion that contribution of glial cells should be considered when interpreting our results (p. 20).

Reviewer 2

In the manuscript entitled "Aberrant activity of mitochondrial NCLX is linked to impaired synaptic transmission and is associated with mental retardation" by Alexandra Stavsky and colleagues describe neuronal alterations resulting from a disease-related mutation in the mitochondrial sodium/lithium/calcium exchanger (NCLX). The main finding is that the P367S mutation of NCLX leads to slowed calcium efflux from mitochondria which results in calcium accumulation in mitochondria and defective buffering of axonal calcium. As consequence, there are reduced basal levels and activity-dependent increase of calcium in neurons. Ultimately, these alterations cause reduced evoked and spontaneous release of neurotransmitter (glutamate) and impair the induction of LTP. I want to congratulate the authors for a well written article which I enjoyed reading, a very detailed method section (uncommon nowadays but necessary for reproducibility) and overall and interesting manuscript. I think that the data presented in the figures, using different techniques (molecular biology, electrophysiology, imaging) and experimental settings (cultures and slices), fully support the authors' conclusions and no further experiments are needed. I only have minor comments that require changes/additions in the text.

We thank the reviewer for her/his enthusiastic support. We are happy that the reviewer finds our manuscript convincing and well-written! We agree that detailed materials and methods sections are important, but are all too rare nowadays.

1. In the Discussion section the authors state that the "results indicate that deletion of NCLX affects not only mitochondrial calcium efflux, but also influx". However, in Figure 1E calcium influx into the mitochondria seems to be similar between WT and NCLX-P367S cells, contradicting the previously mentioned conclusion. How do the authors reconcile this? I think there is no direct evidence in the manuscript for calcium influx alterations.

The statement that the reviewer pointed out does not referring to Figure 1E, which is data from mitochondria in SH-SY5Y cells overexpressing NCLX, but rather to Figure 2F-H, which is data obtained from mitochondrial transients in NCLX KO neurons. We now specifically indicate which information this statement is based on in the discussion (p. 18). We believe that the data presented in Figure 2F-H indeed support an effect of NCLX deletion on calcium influx in neuronal cells. Specifically, we observed that basal calcium is elevated in NCLX KO neurons and that mitochondria are depolarized, which could indeed lead to a reduction in the driving force for calcium influx. We note that in answering the comments of reviewer 1, we redid the whole set of experiments on SH-SY5Y cells, and influx appears to be reduced in this set of results as well (see Figure 1E and Supplemental Figure 1).

2. Regarding the changes in neurotransmission, the authors show a decrease in neurotransmitter release (using pHluorin and also estimating PR via electrophysiology). There is also a decrease in the amplitude of miniature postsynaptic currents, indicating putative changes in postsynaptic receptors but this is not discussed by the authors, can the authors speculate on the meaning of this result? Are postsynaptic receptors reduced in NCLX-P367S neurons? As a side note, there is no indication in the manuscript if the mEPSC recorded are AMPAR- or NMDAR-mediated (the traces look like AMPAR-mediated), this has to be informed.

Indeed, in the present manuscript we do not determine the specific reason for the reduction in the mEPSC amplitude. We agree this is an important and interesting topic, but we feel this should be the topic of follow-up research. Nevertheless, we added to the discussion section

a paragraph describing possible explanations for the reduction in the mEPSC amplitude (p. 19).

To answer the reviewer's specific question concerning the type of receptors involved in the mEPSC recordings, we point out that these were performed in voltage-clamp mode at a holding potential of -70mV, using cesium-based intracellular solution which extends the space clamp of the recordings. Also, 2mM magnesium concentrations in the extracellular solution. Therefore, we assume that most of the mEPSCs we observed were indeed mediated by AMPA-receptors. Due to this question, the technical details of the recordings have been expanded in the methods and materials sections (p. 12), because important details were indeed missing. This should allow informed readers to reach this conclusion.

3. Finally, the authors center the discussion on calcium regulation by mitochondria and its impact on neurotransmission. However, the observed calcium accumulation in mitochondria, taken together with the depolarization of the mitochondrial membrane, can also lead to impairment on ATP production and also generation of reactive species (and the consequent oxidative stress), both processes that can negatively impact on neuronal health and neurotransmission. These alternative explanations need to be added and discussed in the manuscript. As a small note related to this topic, the authors calculate a depolarization of the mitochondrial membrane of ~23 mV, is this a significant change? The meaning of this magnitude (i.e. if it is a big enough change to alter mitochondria metabolism) should be explained.

We added to the discussion (p. 20-21) a paragraph indicating that deletion of NCLX may affect not only the calcium-handling properties of the mitochondria, but also its role as a main provider of cellular energy, as a hub for the synthesis of cellular building blocks and as a potential source of reactive species. In answer to the reviewer's question concerning the significance of a 23mV depolarization of the mitochondria, we discuss that it can affect both calcium handling and ATP generation. However, we cite previous literature showing that similar differences in mitochondrial polarization have been observed within the same mitochondrial network in various tissues, even under physiological settings. We discuss that because neurons, when required, were shown to use glycolysis for ATP production when performing synaptic transmission, the impact of the depolarization of the mitochondria on calcium handling and on ATP production may differ.

REVIEWERS' COMMENTS:

Reviewer #1 (Remarks to the Author):

I want to thank the authors for addressing my questions and comments and commend them on their revised manuscript. I have no further comments.

Reviewer #2 (Remarks to the Author):

The authors have addressed all the points raised by the Reviewers and I think the manuscript is ready for publication.